# Effect of power training on physical functional performance of patients with Parkinson's disease: A systematic review and meta-analysis of randomized controlled trials

Samuel Brito de Almeida[1], Emmanuelle Silva Tavares Sobreira[1], Charles Phillipe de Lucena Alves[2], Danielle Pessoa Lima[1,3], Janine de Carvalho Bonfadini[1], Manoel Alves Sobreira-Neto[4], Thiago Holanda Freitas[5], Simony Lira do Nascimento[6], Pedro Braga-Neto[4,7]*

1 Clinical Research Unit, Hospital Universitário Walter Cantídio, Universidade Federal do Ceará, Fortaleza, Ceara, Brazil, 2 Graduate Program in Epidemiology, University of Pelotas, Pelotas, Rio Grande do Sul, Brazil, 3 Medical School of Universidade de Fortaleza, Universidade de Fortaleza, Fortaleza, Ceara, Brazil, 4 Division of Neurology, Federal University of Ceará, Fortaleza, Ceara, Brazil, 5 Department of Psychiatry, Federal University of São Paulo, São Paulo, Brazil, 6 Department of Physical Therapy, Federal University of Ceara, Fortaleza, Ceará, Brazil, 7 Center of Health Sciences, State University of Ceará, Fortaleza, Ceará, Brazil

* pbraganeto@ufc.br

## Abstract

### Introduction

Parkinson's disease (PD) is becoming more prevalent, highlighting the urgency of developing treatments to minimize its effects on muscular strength and physical function. Power training (PT) is a potential approach that may improve endurance and muscular power, essential for maintaining functional ability in PD.

### Objective

To compare the effect of PT versus control or other physical activity (PA) interventions on physical functional performance (PFP) in PD patients.

### Methods

We searched PubMed, MEDLINE, Embase, LILACS, PEDro, Cochrane Library, and Scopus. Inclusion criteria were randomized controlled trials comparing PT to a control group or another PA intervention in PD patients. PFP was the primary outcome. Pooled effect estimates were calculated from baseline to endpoint scores.

### Results

From 21,558 results, four studies were included in the meta-analysis due to their moderate to high methodological quality. PT showed no significant effect on PFP outcomes compared to control groups (TUG: ES, −0.281; 95% CI, −0.693 to 0.130; P = 0.180; I2:0%; PWS: ES, 0.748; 95% CI, −0.768 to 2.265; P = 0.333; I2:88%; FWS: ES, 0.420; 95% CI, −0.950 to

**Data Availability Statement:** All relevant data are within the manuscript and its Supporting information files.

**Funding:** The author(s) received no specific funding for this work.

**Competing interests:** The authors have declared that no competing interests exist.

1.791; P = 0.548; I2:83%; SLS: ES, 0.161; 95% CI, −0.332 to 0.655; P = 0.521; I2:0%). No differences were found between PT and alternative interventions (TUG: ES, 0.132; 95% CI, −0.394 to 0.657; P = 0.623; I2:0%; BBA: ES, 0.057; 95% CI, −0.430 to 0.544; P = 0.820; I2:0%).

## Conclusion

PT did not improve PFP compared to control or alternative interventions. More studies are needed to explore PT effects (e.g., higher volume, intensity, and combined types) in PD patients.

## Introduction

Parkinson's disease (PD) is a rapidly increasing neurological disorder in the world affecting more than 10 million people worldwide [1,2]. PD is commonly associated with impairment in an individual's physical performance and skeletal muscle force-generating capacity. Muscle contraction might be difficult for who have PD, reducing muscular strength [3,4]. The loss of strength, postural deficits and decreased mobility in PD patients has an impact on their quality of life [5,6]. Given these reasons, common activities such as getting up from a chair and daily life activities could become difficult [4].

Muscle power is positively associated with the ability of older adults to perform activities of daily living. Additionally, muscle power is considered a better predictor of muscle function than strength [7]. Moreover, muscle power shows a faster decline associated with aging than muscle strength and endurance [8,9].

A specific type of resistance training called power training (PT) has grown in popularity as an effective exercise intervention for improving strength, power, and physical functional performance (PFP) in older adults [10,11]. Due to physiological adaptations (i.e. higher frequency motor units), PT might help with muscular power and endurance, both of which are important for functional capacity, especially for patients with PD.

PFP refers to an individual's ability to perform tasks that require physical movement and strength, which are essential for independent living and quality of life, particularly in populations with functional impairments such as Parkinson's Disease (PD). PFP encompasses various domains, including strength, balance, mobility, and endurance, each of which contributes to an individual's overall functional capacity. In our study, PFP is specifically operationalized through a series of standardized mobility tests that are widely recognized in the literature for assessing key aspects of physical function in PD patients, including the Five Times Sit-To-Stand (FTSTS), 30-Second Chair Stand, Backward Walking, Timed Up and Go (TUG), TUG Dual-Task, and the Short Physical Performance Battery (SPPB). These tests collectively provide a multifaceted evaluation of PFP, capturing the diverse aspects of physical function that are often compromised in PD patients [4–6].

In this context, previous studies have pointed out that PT may be able to improve gait velocity [12], balance [12,13] and mobility [12,13] in these patients. On the other hand, other studies have shown that PT failed to improve gait velocity [12] and mobility [14,15] of PD patients. Due to this heterogeneity in findings from previous studies, it is not clear how PT affects physical functional performance. In reviewing the literature, we identified a range of exercise protocols under the umbrella of PT, including variations in equipment, intensity, and duration, leading to seemingly contradictory findings. For example, while some studies reported

improvements in gait velocity, balance, and mobility with PT [12,13], others did not observe significant gains in these same outcomes [14,15]. These discrepancies are not merely conflicting results but rather reflect the variability in exercise prescription across studies, which complicates the interpretation of PT's effectiveness in this population. Considering that an increase in PFP outcomes is associated with a decrease in cardiovascular risk [16–18], it is clinically important to examine the effect of PT on PFP in this population.

The primary gap in the current literature is the lack of a clear consensus on the optimal PT protocol for improving PFP in PD patients, considering the diversity in exercise modalities and dosing. Our systematic review and meta-analysis aim to address this gap by synthesizing evidence from randomized controlled trials (RCTs) to clarify the overall impact of PT on PFP in PD patients, while accounting for variations in exercise interventions.

Therefore, this systematic review and meta-analysis aimed to compare the effect of PT versus control or another physical activity (PA) intervention to improve physical functional performance in patients with PD.

## Materials and methods

This systematic review and meta-analysis followed the Preferred Reporting Items for Systematic Reviews and Meta-Analyses (PRISMA) guidelines (S1 Table) [19]. This systematic review and meta-analysis was prospectively registered in the International prospective register of systematic reviews (PROSPERO), CRD42019139446.

### Literature search strategy

Systematic searches were conducted in the PubMed, MEDLINE, Embase, LILACS, Physical Therapy Evidence (PEDro), Cochrane Library and Scopus databases with the last update on April 20, 2024. No search restrictions were placed on language, country or date of publication. The following combination of search terms was used: (Parkinson's disease OR Parkinson); NOT experimental Parkinson; AND (exercise OR training, physiotherapy, physical therapy, physical activity, muscle strength, strength training, strength, muscle strength, resistance training, power training, high speed, high velocity, low resistance). The full electronic search strategy is provided in S2 Table.

### Selection and exclusion criteria of the literature

The inclusion criteria were defined based on the participants involved, type of intervention, type of comparison group, outcome of interest and study design (PICOS strategy) [19]. Therefore, the following inclusion criteria were considered: (P) participants with confirmed PD diagnosis fulfilling the London's Brain Databank or Movement Disorders Society criteria at any age or disease stage and community-dwelling people [20]; (I) interventions: power training is defined as a form of resistance training designed to enhance muscle power (force × velocity) by performing exercises with maximal concentric speed. The types of exercises categorized as power training in the included studies typically involved multi-joint movements, such as leg presses, chest presses, and knee extensions, utilizing equipment like free weights, machines, or resistance bands. All programs, regardless of intensity, frequency, or session duration, required participants to lift the weight 'as fast as possible' during the concentric phase. This approach is aimed at optimizing neuromuscular adaptations to improve power output in individuals with Parkinson's disease; (C): For the comparator group, we included studies in which participants had no intervention, usual care for the given study setting, waitlist trials incorporating a placebo-based PA program (sham exercises), another form of PA intervention, or drug therapy. Examples include but are not limited to: balance training,

training of any type of strength manifestation (resistance training, among others), aerobic exercise, Tai Chi, movement strategy training, fall prevention education and home exercise prescription. We excluded trials that compared PT versus nutritional supplementation; (O): PFP was considered as the primary outcome. For this, we considered the following tests:1) mobility tests (Five Times Sit-To-Stand, 30-sec chair stand, Backward walking, Timed Up and Go (TUG), Timed Up and Go dual task and the Short Physical Performance Battery); 2) gait tests (fast and comfortable 10 Meter Walk, 4-Meter Walk, preferred walking speed (PWS) and fast walking speed (FWS), Dynamic Gait Index, Modified Gait Efficacy Scale and 6-min walk); and 3) balance tests (Berg Balance Scale (BBS), single leg stance (SLS), Fullerton Advanced Balance scale, Mini-BESTest, Performance-Oriented Mobility Assessment, Biodex Balance and Activities Specific Balance Confidence Scale).

Data regarding these outcomes should be presented at pre and post moments or the change values between these moments. When necessary, the authors were contacted for data recovery; finally, (S): The following study designs were considered: randomized controlled trials (RCTs) and quasi-randomized controlled trials.

All records were assessed for eligibility by two independent reviewers (SB, ES). The reviewers independently selected articles detected for inclusion using the protocol registered and made decisions on inclusions according to the pre-established eligibility criteria. Abstracts which did not adequately describe the inclusion and exclusion criteria were retrieved for full-text review. The reviewers independently evaluated full-text articles and determined study eligibility. Discrepancies were resolved by consensus and by a third author (PBN). Reasons for exclusion of studies were logged throughout the process. In the case of articles with incomplete and potentially eligible data, we tried to contact the authors to request additional data if necessary. The number of articles included and excluded at the different steps was recorded and presented in a PRISMA flowchart (Fig 1).

### Data extraction and quality assessment

Data extraction was performed by the same two independent reviewers (SB and SL) using a data extraction form tailored to the requirements of this systematic review. Any disagreement and/or discrepancy between the two reviewers was resolved through discussion with a third member of the review team (PBN). Some relevant information for systematic reviews was extracted from each study as follows: (1) report characteristics; (2) participants; (3) the PT program characteristics (duration, series, repetitions, intensity, sessions per week, and muscle groups or exercises); (4) outcome measures; and (5) results. We contacted the corresponding authors when relevant information was not reported.

Data extracted from each of the studies for the meta-analysis included the number of participants per group, the outcome effect measure from study baseline to study endpoint for each group. Means and dispersion or precision measures (standard deviation, standard error or 95% confidence interval (CI)) were extracted for each group. Standard error (SE) was converted to standard deviation (SD) by the equation SD = SE × (n), if SD had not been provided in the original study. The 95% confidence intervals were converted to SD considering the equation (n*(UL-LL)/[2*T.INV 0.05;n-1], where: n is the sample size, UL is the upper limit, LL is the lower limit and T.INV is the function that calculates the left-tailed inverse of the Student's t-test distribution [21].

The quality of the evidence was assessed by Grading of Recommendations Assessment, Development and Evaluation [22]. The GRADE approach suggests initially classification as high-quality studies (score 4) for randomized controlled trials, with further reduction to moderate, low, or very low depending on (1) quality of the original studies; (2) inconsistency of the

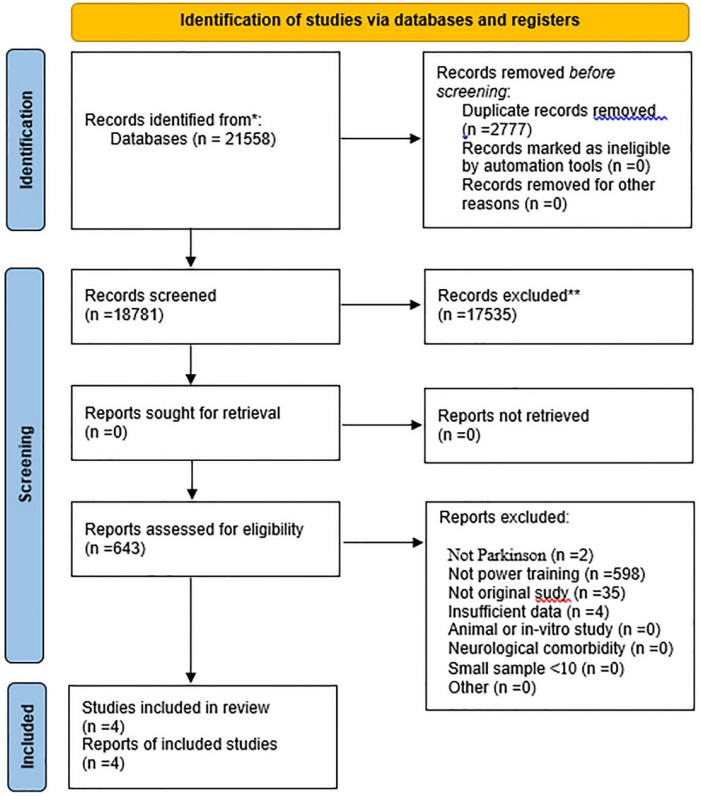

**Fig 1. PRISMA flow diagram of included published studies.**

results (heterogeneity); (3) indirect evidence; (4) imprecision; and (5) publication bias. To determine the quality of the evidence, the following criteria were used: high level of quality research is not likely to undermine our confidence in the effect estimate; moderate quality— further study may modify the estimate and have a significant influence on our confidence in the estimate of effect; Low quality means that we are uncertain about the estimate, very low quality means that it is possible that future research will significantly affect our confidence in the estimate of effect and modify the estimate.

## Risk of bias assessment

The methodological quality of the included trials was evaluated using the Revised Cochrane risk-of-bias tool for randomized trials (RoB 2) [23]. This tool assesses bias across five key domains: 1) bias arising from the randomization process, 2) bias due to deviations from intended interventions, considering both the effect of assignment to the intervention and adherence to it, 3) bias due to missing outcome data, 4) bias in the measurement of the outcome, and 5) bias in the selection of the reported results. Each domain is examined through 3–7 signaling questions, with possible responses including: yes, probably yes, probably no, no, and no information. Based on the responses to these questions, an algorithm determines the risk of bias for each domain, categorizing it as "low risk of bias," "high risk of bias," or "some concerns."

After evaluating each domain individually, an overall risk of bias judgment for each study was made using the same classification criteria. A study was deemed to have a "low risk of

bias" if all domains met the required standards. If at least one domain raised "some concerns" or lacked clarity, the study was classified under "some concerns." A "high risk of bias" designation was assigned when at least one domain did not meet the standards, or when multiple domains were classified as having "some concerns." Two independent reviewers (SB and DP) assessed the risk of bias of the included studies, and any disagreements in the scoring of trials were resolved with a third reviewer.

## Data synthesis and statistical analyses

Pooled effect estimates were calculated from the scores of changes between baseline and endpoint of interventions, their standard deviations (or 95% confidence intervals), and the number of participants in each group.

The Hedges'g effect sizes were presented as standardized mean differences (SMD—a measure of effect, recommended to be used when the study reports the effectiveness of an intervention in continuous measurements, especially in cases of different measurement methods) and calculations were performed using random-effect models, applying the DerSimonian-Laird method. Hedges' g includes a correction for small sample size bias and is often recommended in meta-analyses because correcting for small sample sizes results in a more conservative and precise estimate, especially when the meta-analysis includes studies with varying sample sizes [24].

As some studies [13,15] compared the PT data with two comparator groups (each study), this shared PT data group was divided into two groups with smaller sample sizes weighted in relation to different comparator groups. This approach was applied to have reasonably independent comparisons and overcome a unit-of-analysis error for studies that could contribute to multiple and correlated comparisons, as recommended by the Cochrane Handbook for Systematic Reviews of Interventions [21].

Statistical heterogeneity was evaluated by Cochran's Q statistic and the $I^2$ inconsistency test. Values > 50% are considered indicators of high heterogeneity. Forest plots were generated to show the combined effect and SMD with 95% confidence intervals (CIs). Statistical significance was established with a p-value <0.05. The meta-analyses were performed using Open-Meta Analyst software program [25].

## Results

The literature search identified a total of 21,558 results. Of those which 2,777 duplicates were removed, and 17,535 studies were excluded according to title and abstract. Thus, a total of 643 unique full-text articles were assessed for eligibility. Of these, 639 studies were excluded because they did not meet the eligibility criteria. The titles and reasons for exclusion are available in the (S3 Table). After reviewing the full text reading, 4 studies were selected for the meta-analysis (Fig 1) [12–15].

## Participants

Table 1 summarizes the main results of the included studies. A total of 158 participants (women, n = 56; men, n = 102) with a median age of 70.2 years (interquartile range [IQR] = 65.0 to 73.0 yrs) were included in this review. Two studies were conducted in United States of America [13,14], one in Australia [12] and one in Belgium [15]. All included studies [12–15] recruited participants with mild to moderate PD (Hoehn and Yahr stage I to III). Disease duration, mean Mini Mental State Examination (MMSE), and Movement Disorder Society Unified Parkinson's Disease Rating Scale Part III (MDS-UPDRS part III) showed variations among the included studies. All studies [12–15] used anti-Parkinsonian medications as usual therapies

**Table 1. Summary of the data regarding the characteristics of the study sample included in the systematic review.**

| Article authors and year | Study design | Participants | | | | | | Outcomes | PWT Protocol intervention | | | | Control group Intervention | | | | Third group: Control group Intervention |
|---|---|---|---|---|---|---|---|---|---|---|---|---|---|---|---|---|---|
| | | Sample Size, N | Mean age (SD) | H&Y | Mean years since diagnosis (SD) | Mean MMSE (SD)—Pre | UPDRS—Part III—Pre | | Number of exercises (ex); parts of the body | Intensity; number of sets and repetitions | Weekly frequency; | Type of exercises | Number of exercises (ex); parts of the body | Intensity; number of sets and repetitions/ duration | Weekly frequency; intervention duration(weeks) | | |
| Cherup et al., 2019 [14] | RCT | PWT group 21 | PWT group 73.0 (6.8) | 1–3 | No data | PWT group 28.3 (1.7) | PWT group 32.8 (11.6) | Strength (LP, CP); Power (LP,CP); DMA; BBA; TIME | 10 ex; whole body. | 30%–50% 1RM; 3x 10 Rep. | 2x/week; 12 weeks. | Strength training | 10 ex; whole body. | 30%–70% 1RM; 3x 10 Rep. | 2x/week; 12 weeks. | | Not reported |
| | | ST group 21 | ST group 69.3 (10.5) | | | ST group 29.0 (1.5) | ST group 28.6 (16.2) | TUG; MFES; PDQ-39. | | | | | | | | | |
| Demonceau et al., 2017 [15] | 3 arm RCT | ST group 17 | ST group 75.0 (10.0) | 1–3 | ST group 7.0 (2–9) | ST group 28 (26–30) | ST group 20.0 (7.8) | Strength (KE, FM); VO2peak; PWL; RER; HRR; Gait analysis (speed, stride length and cadence); TUG; 6MWD; PASS; PDQ-39. | Not reported; whole body. | 50%–90% 1RM; 2-3x 05–15 Rep. | 2-3x/week; 12 weeks. | Aerobic training | NA | 50%–80% PWL; 30–45 Min | 2-3x/week; 12 weeks. | | Standard care program |
| | | AE group 20 | AE group 65.0 (8.0) | | AE group 5 (2.5–8) | AE group 28 (27–29) | AE group 16.9 (6.8) | | | | | | | | | | |
| | | SC group 15 | SC group 63.3 (6.0) | | SC group 5 (3–7) | SC group 28 (27–29) | SC group 16.3 (9.2) | | | | | | | | | | |
| Ni et al., 2016 [13] | 3 arm RCT | PWT group 14 | PWT group 71.6 (6.6) | 1–3 | PWT group 6.6 (4.4) | No data | PWT group 32.9 (12.0) | UPDRS–part III; BBA; MINI-BEST; SLS; PS; FR; TUG. | 11 ex; whole body. | 50%–75% of the optimal loads (one-week adaptation period) + increased weekly; 3x 10–12 Rep. | 2x/week; 12 weeks. | Yoga | NA | NA; 60 Min | 2x/week; 12 weeks. | | Health education classes |
| | | Yoga group 15 | Yoga group 71.2 (6.5) | | Yoga group 6.9 (6.3) | | YOGA group: 28.15(11) | PWS; FWS; Strength LP; Power LP. | | | | | | | | | |
| | | Control group 12 | Control group 74.9 (8.3) | | Control group 5.9 (6.2) | | Control group 27.6 (7.8) | | | | | | | | | | |
| Paul et al., 2014 [12] | RCT | PWT group 20 | PWT group 68.1 (5.6) | 1–3 | PWT group 7.8 (5.2) | PWT group 29.1 (1.4) | PWT group 37.1 (11.0) | Power (KE, FM, HF, HA, SF, EE) | 4 ex; lower body. | 40%–60% 1RM; 3x08 Rep. | 2x/week; 12 weeks. | Strength training | 4 ex; lower body. | Low-intensity;2x8-12 Rep. | 2x/week; 12 weeks. | | Not reported |
| | | Control group 20 | Control group 64.5 (7.4) | | Control group 7.8 (5.9) | Control group 28.9 (1.3) | Control group 35.7 (14.0) | Strength (KE, FM, HF, HA, SF,EE) | | | | | | | | | |
| | | | | | | | | PWS; FWS; TUG | | | | | | | | | |
| | | | | | | | | BALANCE (CSRT, MBR, SLT) | | | | | | | | | |
| | | | | | | | | N-FOG | | | | | | | | | |

Note: AE = Aerobic exercise; BBA = Berg balance assessment score; CP: Chess press; CSRT = Choice stepping reaction time; DMA = Dynamic motion analysis score; EE = Elbow extensors; FM = Flexor muscles; FR = Functional reach; FWS = Fast walking speed; HA = Hip abductors; HF = Hip flexors; HRR = Heart rate reserve; H&Y = Hoehn and Yahr stage; KE = Knee extension; LP = Leg press; MBR = Maximal balance range; MFES = Modified falls efficacy scale; MINI-BEST = Mini-balance evaluation systems; MMSE = Mini Mental State Examination; NA = Not applicable; N-FOG = New freezing of gait; PASS = Physical activity status scale; PDQ-39 = The Parkinson disease questionnaire-39; PS = Postural sway; PWL = Peak work load; PWS = Preferred walking speed; PWT = Power training; RCT = Randomized controlled trial; RER = Respiratory exchange ratio; SC = Standard care; SF = Shoulder flexors; SLS = Single leg stand; ST = Strength training; TIME = Time on platform; TUG = Timed up and go; UPDRS = Unified Parkinson's Disease Rating Scale; VO2peak = Peak oxygen consumption; 1RM = repetition maximum; 6MWD = Six-minute walk distance.

regardless of the intervention and performed the outcome measurements during the clinical ON state of the patients.

## Intervention

The intervention period was 12 weeks for all included studies. The intervention frequency of twice a week was the same for three studies [12–14], while another [15] carried out exercise three times a week. The exercise duration in each session varied from 45 minutes [12] to 90 minutes [15]. The exercise intensity in each program varied from 30% (14) to 90% [15] of 1RM (one-repetition maximum).

## Outcome measures

The TUG test was the most common evaluation mobility outcome, presented in 4 studies [12–15]. The BBS was used for balance evaluation in two studies [13,14], while SLS was also used in two articles [12,13]. Gait velocity was assessed in two articles [12,13] by two methods: 10 m in the preferred speed (m/s) and in the fast speed (m/s). Table 2 provides a detailed description of how different programs have affected various outcomes in several studies. The information includes the mean values for the baseline and endpoints as well as the mean difference with 95% confidence intervals indicating the change that was seen at the 3-month mark. The table mainly presents results for Timed Up and Go (TUG), preferred and rapid walking speeds, and balance-related metrics such as Single-Leg Stance (SLS) and Berg Balance Scale (BBA-BERG). Exercises that are used as interventions include yoga, aerobic, power, and strength training, along with control groups for comparison. While yoga also showed good impacts on balancing outcomes, the data suggest that different interventions were not all that helpful. Power and aerobic training showed the greatest gains in TUG and walking speed.

## Synthesis of the results

**Meta-analysis of TUG.**   Data regarding TUG were available from 3 studies [12,13,15], which compared PT interventions versus control groups or alternative interventions in a total of 175 participants. PT was not associated with changes in TUG time compared with control (ES: −0.28; 95% CI, −0.69 to 0.13; P = 0.18; I2: 0%) (Fig 2), as well as with alternative interventions (ES: 0.13; 95% CI, −0.39 to 0.65; P = 0.62; I2: 0%) (Fig 3).

**Meta-analysis of PWS.**   Data regarding PWS were available from 2 studies [12,13], which evaluated PT interventions versus control groups in a total of 81 participants (Fig 4). PT was not associated with changes in PWS time compared with control (ES: 0.74; 95% CI, −0.76 to 2.26; P = 0.33; I2: 88%).

**Meta-analysis of FWS.**   Data regarding FWS were available from 2 studies [12,13], which compared PT interventions versus control groups in a total of 81 participants (Fig 5). PT was not associated with changes in FWS time compared with control (ES: 0.42; 95% CI, −0.95 to 1.79; P = 0.54; I2: 83%).

**Meta-analysis of BBS.**   Data regarding BBS were available from 2 studies [13,14], which assessed PT interventions versus alternative intervention in a total of 83 participants (Fig 6). PT was not associated with changes in BBS score compared with other interventions (ES: 0.05; 95% CI, −0.43 to 0.54; P = 0.82; I2: 0%).

**Meta-analysis of SLS.**   Data regarding SLS were available from 2 studies [12,13], which investigated PT interventions versus control groups in a total of 81 participants (Fig 7). PT was not associated with changes in SLS time compared with control (ES: 0.16; 95% CI, −0.33 to 0.65; P = 0.52; I2: 0%).

**Table 2. Summary of intervention effects on various outcomes.**

| Study | Outcome | Intervention | N | Baseline Mean (SD) | End Point Mean (SD) | Change at 3-Month Time Point (95% CI) |
|---|---|---|---|---|---|---|
| Demonceau et al., 2017 [15] | TUG | Aerobic | 16 | 6.4 (1.64) | 5.87 (1.57) | -0.53 (-1.10 to -0.04) |
| | | Power Training | 14 | 7.54 (3.15) | 6.68 (2.33) | -0.86 (-2.20 to 0.48) |
| | | Control | 15 | 6.42 (1.42) | 6.24 (1.13) | -0.18 (-0.55 to 0.19) |
| Ni et al., 2016 [13] | TUG | Power Training | 14 | 10.8 (5.5) | No data | -1.3 (-2.4 to -0.3) |
| | | Yoga | 13 | 10.27 (3.9) | No data | -2.3 (-4.1 to -0.6) |
| | | Control | 10 | 10.2 (2.4) | No data | 0.3 (-0.3 to 0.9) |
| Paul et al., 2014 [12] | TUG | Power Training | 20 | 9.7 (2.3) | 8.3 (2.4) | No data |
| | | Control | 20 | 9.5 (2.8) | 8.6 (4.3) | No data |
| Cherup et al., 2019 [14] | TUG | Power Training | 17 | 11.1 (12.4) | 9.18 (7.4) | No data |
| | | Strength Training | 18 | 7.5 (2.4) | 7.8 (3.1) | No data |
| Ni et al., 2016 [13] | PWS | Power Training | 16 | 1.03 (0.27) | No data | 0.12 (0.06 to 0.18) |
| | | Yoga | 14 | 1.06 (0.2) | No data | 0.14 (0.02 to 0.26) |
| | | Control | 15 | 1.04 (0.28) | No data | - 0.03 (-0.06 to 0.01) |
| Paul et al., 2014 [12] | PWS | Power Training | 20 | 1.27 (0.17) | 1.34 (0.22) | No data |
| | | Control | 20 | 1.17 (0.31) | 1.24 (0.38) | No data |
| Ni et al., 2016 [13] | FWS | Power Training | 16 | 1.52 (0.42) | No data | 0.16 (0.08 to 0.24) |
| | | Yoga | 14 | 1.49 (0.25) | No data | 0.22 (0.08 to 0.36) |
| | | Control | 15 | 1.41 (0.43) | No data | 0.002 (-0.06 to 0.06) |
| Paul et al., 2014 [12] | FWS | Power Training | 20 | 1.77 (0.25) | 1.81 (0.31) | No data |
| | | Control | 20 | 1.67 (0.39) | 1.7 (0.44) | No data |
| Ni et al., 2016 [13] | BBA—BERG | Power Training | 14 | 48.8 (5.8) | No data | 4.4 (2.9 to 6.0) |
| | | Yoga | 13 | 49.22 (3.9) | No data | 4.2 (2.4 to 5.9) |
| | | Control | 10 | 50.9 (6.1) | No data | 0.4 (-0.1 to 0.9) |
| Cherup et al., 2019 [14] | BBA—BERG | Power Training | 17 | 50.88 (5.67) | 49.47 (7.12) | No data |
| | | Strength Training | 18 | 50.94 (7.04) | 49 (6.82) | No data |
| Paul et al., 2014 [12] | SLS (Right) | Power Training | 20 | 14.64 (10.63) | 18.59 (15.91) | No data |
| | | Control | 20 | 21.36 (19.18) | 22.32 (19.35) | No data |
| Paul et al., 2014 [12] | SLS (Left) | Power Training | 20 | 11.17 (5.94) | 13.63 (10.26) | No data |
| | | Control | 20 | 19.85 (16.6) | 19.78 (19.69) | No data |
| Ni et al., 2016 [13] | SLS (More Affected) | Power Training | 14 | 4.2 (4.1) | No data | 8.4 (- 2.8 to 19.6) |
| | | Yoga | 13 | 6.11 (2.6) | No data | 5.9 (1.7 to 10.2) |
| | | Control | 10 | 10.5 (8.2) | No data | 4.1 (-4.0 to 12.2) |

*(Continued)*

**Table 2.** (Continued)

| Study | Outcome | Intervention | N | Baseline Mean (SD) | End Point Mean (SD) | Change at 3-Month Time Point (95% CI) |
|---|---|---|---|---|---|---|
| Ni et al., 2016 [13] | SLS (Less Affected) | Power Training | 14 | 4.8 (3.6) | No data | 11.8 (2.5 to 21.0) |
| | | Yoga | 13 | 9.24 (7.4) | No data | 17.7 (-0.3 to 35.8) |
| | | Control | 10 | 15.1 (25.6) | No data | 3.4 (- 5.4 to 12.2) |

Note: TUG = Timed up and go; PWS = Preferred walking speed; FWS = Fast walking speed; BBA = Berg balance assessment score; SLS = Single leg stand; SD = standard deviation; CI = Confidence interval.

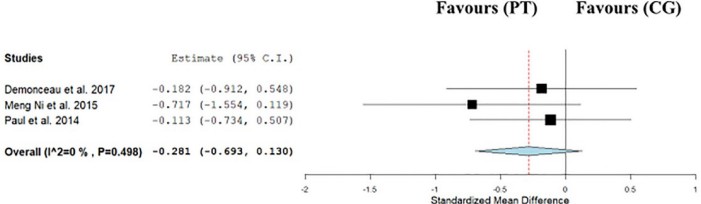

**Fig 2. Forest plot of the comparison between Power training (PT) and control group (CG) effects on timed up and go (TUG).**

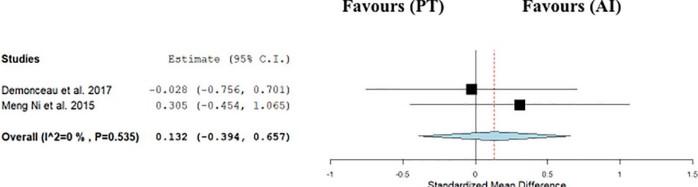

**Fig 3. Forest plot of the comparison between Power training (PT) and alternative interventions (AI) effects on timed up and go (TUG).**

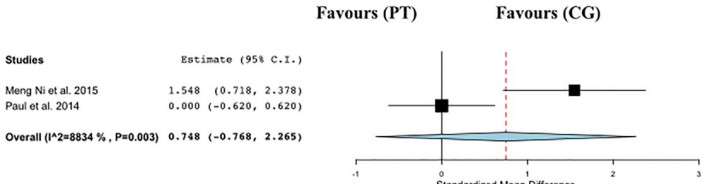

**Fig 4. Forest plot of the comparison between Power training (PT) and control group (CG) effects on preferred walking speed (PWS).**

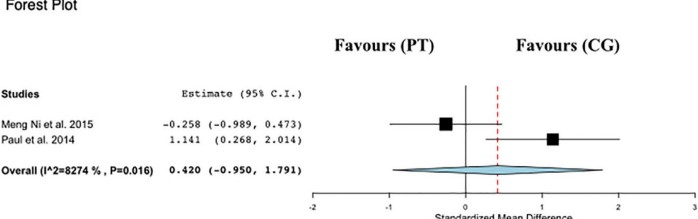

**Fig 5. Forest plot of the comparison between Power training (PT) and control group (CG) effects on fast walking speed (FWS).**

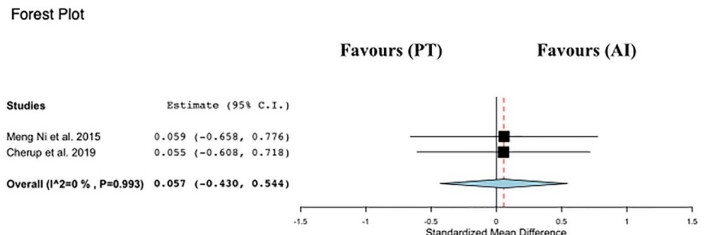

**Fig 6. Forest plot of the comparison between Power training (PT) and alternative interventions (AI) effects on preferred walking speed (BBS).**

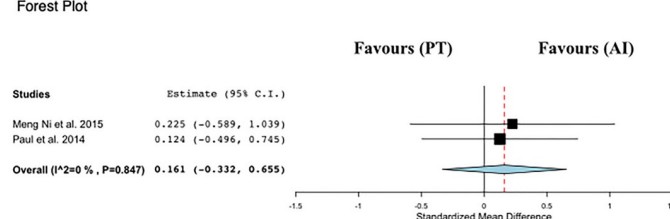

**Fig 7. Forest plot of the comparison between Power training (PT) and alternative interventions (AI) effects on preferred walking speed (BBS).**

It was not possible to conduct a meta-analysis comparing PT to an alternative intervention for the PWS, FWS, SLS outcomes and comparing PT to control groups for the BBS because there was only one trial [13].

## Quality of evidence (GRADE)

The overall certainty of evidence across studies of the outcomes was performed in Table 3. After evaluating the studies using the GRADE criteria, we found that the level of evidence was generally low to very low. This is primarily due to several factors: high risk of bias, inconsistency in the results, and serious imprecision due to the small sample sizes in the included studies.

Specifically, for the TUG test, comparing the PWT group with the control and alternative interventions, the evidence was rated as low due to a very serious risk of bias and serious imprecision, despite no significant inconsistency ($I^2 = 0\%$). For physical performance

**Table 3. GRADE approach: Evidence profile for meta-analysis.**

| Subgroup | Nº of studies | Study design | Risk of bias | Inconsistency | Indirectness | Imprecision | Other considerations | | Quality of evidence |
|---|---|---|---|---|---|---|---|---|---|
| TUG-PWT X CG | 3 | RCT | Very Serious[a] | Not Serious (I2 = 0%) | Not serious | Serious [b] | Serious[b] | None | Low |
| TUG-PWT X AI | 2 | RCT | Serious[a] | Not Serious (I2 = 0%) | Not serious | Serious [b] | Serious[b] | None | Low |
| PWS-PWT X CG | 2 | RCT | Serious[a] | Serious (I2 = 88%) | Not serious | Serious [b] | Serious[b] | None | Very Low |
| FWS-PWT X CG | 2 | RCT | Serious[a] | Serious (I2 = 82%) | Not serious | Serious [b] | Serious[b] | None | Very Low |
| BBS-PWT X AI | 2 | RCT | Serious[a] | Not serious (I2 = 0%) | Not serious | Serious [b] | Serious[b] | None | Low |
| SLS–PWT X AI | 2 | RCT | Serious[a] | Not serious (I2 = 0%) | Not serious | Serious [b] | Serious[b] | None | Low |

TUG: Timed Up and Go; PWT: Power Training; PWS: Preferred Walking Speed; FWS: Fast Walking Speed; BBS: Berg Balance Scale; SLS: Single Leg Stance; RCT: Randomized Clinical Trial; CG: Control group; AI: Alternative interventions.

[a] Risk of bias detected.

[b] Do not have optimal information size.

measures such as PWS and FWS, the certainty of evidence was low, with concerns including serious inconsistency ($I^2$ = 88% and 82%, respectively) and imprecision.

Similarly, balance measures like the BBS and SLS also demonstrated a low level of evidence, primarily due to serious risks of bias and imprecision. The overall assessment highlights that while these studies provide some insight into the effects of power training in Parkinson's Disease, the evidence is not strong, necessitating further high-quality research to draw more definitive conclusions.

## Risk of bias

The summary of the risk of bias analysis is depicted in Fig 8, according to the RoB2 tool [23]. Overall, the studies exhibited varying levels of risk. Two studies [12,14] demonstrated a high risk of bias across multiple domains, particularly in domains related to randomization and measurement of the outcome (D1 and D3). One study [15] showed some concerns, primarily in domain 5, related to the appropriateness of the analysis considering the missing data and deviations from the intended intervention. Another study [13] displayed a mix of low risk in randomization but concerns or high risks in other areas, especially in domain 4, which relates to the influence of missing data on the outcomes.

The high risk in domain 1 for some studies [12,15] was attributed to the lack of clarity on the randomization process. Similarly, domain 3 had concerns due to potential deviations from the intended interventions that were not accounted for in the analysis [12,14]. Domain 5, concerning selective reporting of outcomes, also contributed to the overall concerns as it was unclear whether the outcomes were analyzed as per the pre-specified plans [13]. These issues

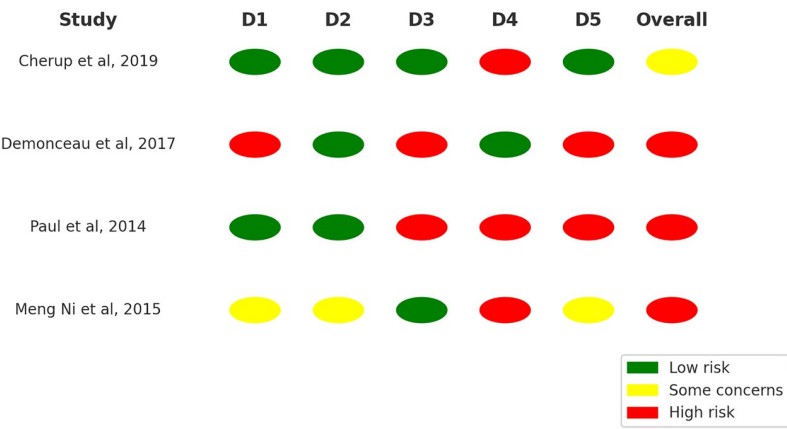

**Fig 8. Summary of risk of bias by domain and study.** D: Domain; D1: Randomisation process; D2: Deviations from the intended interventions; D3: Missing outcome data; D4: Measurement of the outcome; D5: Selection of the reported result.

collectively led to an overall categorization of high risk for some studies and some concerns for others.

## Discussion

This systematic review and meta-analysis aimed to investigate the efficacy of PT in comparison to a control group or other PA intervention on improving physical functional performance in individuals with PD. The main findings were PT interventions had no superior effect on PFP outcomes in PD patients when compared to control groups or alternative interventions.

The results of the present study may partially be explained by some factors. First, some studies showed high heterogeneity–ranging from 88% for PWS and 83% for FWS. There are considerable differences related to the training program between the studies such as (1) intensity; (2) number of sets; (3) repetitions; (4) number of exercises; (5) different parts of the body; (6) duration in each session. Such differences in the study settings may hinder summarizing their results.

Second, the training loads used for muscular power training were lower than the loads used in traditional resistance training programs [12,13]. When conducted at an appropriate intensity through periodization, PT exercises show positive neuromuscular changes in both healthy older adults and in those with chronic conditions [26,27]. The lower loads may also be the reason for the non-superior effects on gait speed-based tests that we found. Some prior studies have been conducted to investigate the ideal load for training power muscles in various populations [10,28,29]. Although there is no common understanding of the most efficient technique or optimal load for power generation [11,30,31], some studies have shown that a resistance training protocol employing power training patterns produces a trend of more power than controlled speed training, notably at moderate and high loads [26,32]. In this sense, a study by de Vos et al. [26] showed that the proportional contribution of force to peak power generation increased with moderate- to high-intensity training.

Third, three of the four included studies [13–15] had their prescribed load intensity determined using the one-repetition maximum, which represents the maximum weight a person can lift for a single repetition throughout the full range of motion [12,33]. A simple and efficient alternative to prescribing the load intensity for PT is through the load-power relationship

[33]. The load-power relationship enables identifying the optimal power generation load in ranges of submaximal loads capable of generating a high amount of mechanical power [33].

There is extensive literature that uses training programs with ideal power load, identifying the highest power at different intensities in addition to considering the ideal power load for each exercise used during PT [31,34,35]. In this sense, only one of the four included studies [13] used optimal load as an intensity parameter, which may have influenced our results. Thus, we endorse more research comparing various intensities and addressing the force- and speed-dependent aspects of power adaptations [30]. Well-designed research utilizing standardized procedures are required in order to examine the effects of PT conducted with various loads on individuals with PD. In a network meta-analysis of 79 trials that summarized the topic, Lopez et al. (2023) [36] concluded that prescriptions based on the velocity of contraction should be individualized to each participant's specific functional needs [36]. We believe that each individualized training can optimize the increase in power, increase training effectiveness and improve the motor symptoms of Parkinson's disease.

Fourth, participants' variations in baseline muscular power were not taken into consideration in the included trials. Paul et al. [12] included participants with a higher mean score of UPDRSIII and higher mean disease duration compared to the participants in the study by Ni et al. [13]. These variables are important determinants of lower muscle power [37]. Previous studies have demonstrated that there is a significant correlation between UPDRS III and muscle activity [37,38]. Given that lower muscle activity indicates a lower number of recruited and de-recruited motor units and has a smaller range of activity [39], the difference among UPDRS III scores from the baseline may have had an impact on our results. In addition, Paul et al. [12] included older adults from a geriatric care setting where muscle power is likely lower than in community-dwelling older adults.

Fifth, regarding the types of exercises, Ni et al. [13] provided high-speed training using 11 pneumatic machine exercises. The results showed improvement in balance and mobility probably due to the specific balance and agility exercise combined with PT. Previous studies [40,41] increased the motor complexity of the exercises. For instance, Silva-Batista et al. [40] investigated the effects of resistance training in both stable and unstable settings (e.g., balance pad, dyna discs, balance discs, BOSU®, and Swiss ball). Although they reported similar increases in leg strength with resistance training in both settings, only the RT instability group had improvements in the TUG and MDS-UPDRS-III scores. This supports the idea that resistance training should be used in conjunction with needs-specific challenges to optimize functional improvements [40,41].

Furthermore, as Paolucci et al. [42] described, PRT along with balance training should be incorporated in rehabilitation programs for PD patients for postural control and other forms of exercise to maintain cardiorespiratory fitness and increase endurance in daily living activities. Considering that muscle power (force × velocity) [43] has positive effects on strength [12,13,15], mobility [13,15] and balance [12,13] in PD populations, and that bradykinesia is the main issue involved in the functionality of PD patients [13], this is a critical insight which demonstrates the importance of PT plus different exercise training modalities in PD treatment.

The results of a recent systematic review and meta-analysis [44] conducted to learn more about how regular PA affects motor symptoms in people with early and intermediate stages of Parkinson's disease (PD), showed that PA interventions have a positive impact on motor symptoms, gait function, and balance in those who are in the early and mid-stages of the disease. The study's findings support the use of PA as a non-pharmacological treatment option for those with early- and mid-stage PD. However, only one of the 15 included studies used PT,

which emphasizes the need for more clinical trials to determine the advantages of PT for PD patients.

Finally, our systematic review included studies with intervention periods of only 12 weeks. Furthermore, the weekly frequency was twice for three studies [12–14] and three times for one study [15]. These factors may also be the reason for the smaller effects found on PFP outcomes. In a network meta-analysis of 250 studies involving 13,011 participants with PD, Yang et al. [45] showed that exercise duration and weekly frequency might significantly alter the efficacy of workouts on the improvement of motor symptoms, suggesting that it is more effective when conducted for over 24 weeks and five times per week. While our discussion highlighted several factors that may have compromised the meta-analysis, it is important to also focus on the potential clinical implications of the results. The consistent improvements seen in key parameters, such as TUG and BBA, suggest that PT can have significant benefits in improving PFP in patients with PD. These findings are particularly relevant in enhancing mobility and balance, which are crucial for maintaining independence in daily activities [46]. The variability in exercise dosages across the included studies may reflect real-world differences in clinical practice, yet the consistent positive outcomes observed in some measures underscore the adaptability and effectiveness of PT interventions. Therefore, while methodological limitations exist, the results of this meta-analysis contribute valuable insights into the role of PT as a therapeutic strategy in PD, demonstrating its potential to mitigate functional decline and improve quality of life.

## Limitations

Additional research should concentrate on randomized trials with greater sample sizes beyond the conclusion of the intervention to enable definitive therapeutic advice. The limitations of this systematic review are typical given the nature of the study selection. First, only one study [13] used the MDS-UPDRS motor scale as an outcome. Given that the MDS-UPDRS motor battery is the gold standard to assess PD motor progression and response to treatments [47], this scale should be used in all studies.

Secondly, high heterogeneity was indeed observed in two meta-analyses among the six conducted (related to PWS and FWS outcomes). In the remaining four meta-analyses, heterogeneity was 0%. We understand that high heterogeneity represents an opportunity to explore potential differences among the included studies. In this context, we sought to assess the feasibility of conducting sensitivity analyses and meta-regressions. However, in attempting to meet the prerequisites for such analyses, we found that it could not be performed. According to the Cochrane Handbook, meta-regressions are viable only when there are at least 10 studies/comparisons for each outcome [21]. In both analyses (where heterogeneity was high), we had only 2 studies included, which precludes the possibility of performing sensitivity analyses and meta-regressions due to limited literature on the topic. Further, in an effort to understand the heterogeneity, we qualitatively analyzed the two included studies, particularly regarding their samples and methodologies. Upon careful review, we observed that the study by Ni et al. (2016) [13] included an older sample with less severe disease symptoms (UPDRS-III) compared to the study by Paul et al. (2014) [12]. Additionally, the training protocols differ somewhat (Ni's protocol included more exercises and greater intensity compared to Paul's). We believe these differences between the studies may have led to the varying adaptations in PWS and FWS observed, resulting in high heterogeneity. However, it is important to note that both studies meet the eligibility criteria for this review and adhere to the established PICOT criteria. Additionally, we have utilized random-effects models and applied appropriate statistical techniques to account for heterogeneity when pooling data.

Thirdly, it was not possible to perform a meta-regression due to the small number of studies preventing a subgroup analysis according to type of intervention. Future power training intervention studies should use the TIDieR framework to fully describe the complexity of the intervention [48]. Fourth, Grey literature was not included in the search [49]. As a strength, this was the first study designed and conducted to summarize the effects of PT compared with a control group or another PA intervention in people with PD. This review also identified critical research topics that must be addressed to fully comprehend the consequences for clinical practice. Future research should: 1) develop a standard evaluation for assessing PFP that can produce consistent findings; and 2) explore and compare different intensities (i.e. low x heavy).

## Conclusions

PT did not have superior improvement PFP when compared to a control group or alternative interventions. Due to the high heterogeneity and the small number of studies, we suggest that our findings should be interpreted with caution. More studies investigating effects of PT (e.g., higher volume, intensity, frequency and combined types) in people with PD will be necessary to better address this important research question.

## Supporting information

**S1 Table. PRISMA checklist.**
(PDF)

**S2 Table. Search strategy.**
(PDF)

**S3 Table. Excluded studies and reasons.**
(PDF)

## Author Contributions

**Conceptualization:** Samuel Brito de Almeida, Danielle Pessoa Lima, Manoel Alves Sobreira-Neto, Thiago Holanda Freitas, Simony Lira do Nascimento, Pedro Braga-Neto.

**Data curation:** Samuel Brito de Almeida.

**Formal analysis:** Simony Lira do Nascimento.

**Investigation:** Samuel Brito de Almeida, Emmanuelle Silva Tavares Sobreira, Pedro Braga-Neto.

**Methodology:** Samuel Brito de Almeida, Emmanuelle Silva Tavares Sobreira, Manoel Alves Sobreira-Neto, Thiago Holanda Freitas, Simony Lira do Nascimento, Pedro Braga-Neto.

**Project administration:** Pedro Braga-Neto.

**Supervision:** Thiago Holanda Freitas, Simony Lira do Nascimento, Pedro Braga-Neto.

**Writing – original draft:** Samuel Brito de Almeida, Charles Phillipe de Lucena Alves, Danielle Pessoa Lima, Janine de Carvalho Bonfadini, Manoel Alves Sobreira-Neto.

**Writing – review & editing:** Samuel Brito de Almeida, Emmanuelle Silva Tavares Sobreira, Charles Phillipe de Lucena Alves, Danielle Pessoa Lima, Janine de Carvalho Bonfadini, Manoel Alves Sobreira-Neto, Thiago Holanda Freitas, Simony Lira do Nascimento, Pedro Braga-Neto.

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
