## [Decision Letter · Decision Letter 0]

1 Apr 2024

PONE-D-23-42955EFFECT OF POWER TRAINING ON PHYSICAL FUNCTIONAL PERFORMANCE OF PATIENTS WITH PARKINSON’S DISEASE: A SYSTEMATIC REVIEW AND META-ANALYSIS OF RANDOMIZED CONTROLLED TRIALSPLOS ONE

Dear Dr. BRAGA-NETO,

Thank you for submitting your manuscript to PLOS ONE. After careful consideration, we feel that it has merit but does not fully meet PLOS ONE’s publication criteria as it currently stands. Therefore, we invite you to submit a revised version of the manuscript that addresses the points raised during the review process.

We look forward to receiving your revised manuscript.

Kind regards,

Hamid Reza Baradaran, M.D., Ph.D.,

Academic Editor

PLOS ONE

Journal Requirements:

"The author Pedro Braga Neto received funding from the Brazilian National Council for Scientific and Technological Development (CNPq) as research grant funding (Productivity scholarship). The author Samuel Brito de Almeida received funding from Ceará Foundation of Support to Scientific and Technological Development (FUNCAP) as doctoral scholarship. The author Emamnuelle Silva Tavares Sobreira received a research grant funding (Scientific and regional development scholarship) from CNPQ/FUNCAP."

3. Please upload a new copy of Figures 1 to 7 as the detail is not clear. Please follow the link for more information:

https://blogs.plos.org/plos/2019/06/looking-good-tips-for-creating-your-plos-figures-graphics/

https://blogs.plos.org/plos/2019/06/looking-good-tips-for-creating-your-plos-figures-graphics/

5. We note that this manuscript is a systematic review or meta-analysis; our author guidelines therefore require that you use PRISMA guidance to help improve reporting quality of this type of study. Please upload copies of the completed PRISMA checklist separately as Supporting Information with a file name “PRISMA checklist".

**Additional Editor Comments:**

Interesting study ; would suggest

1- Define your study PICO precisely ;

2- Please be sure about the literature review and comprehensive search in all relevant debases;

3- Please employ ROB II in order to evaluate risk of bias as you see in https://methods.cochrane.org/risk-bias-2

4- Apparently seems the results of this systematic review has high heterogeneity . Heterogeneity is an important consideration in systematic reviews, as high heterogeneity may imply that it is not suitable to perform meta-analysis

Reviewers' comments:

Reviewer's Responses to Questions

**Comments to the Author**

1. Is the manuscript technically sound, and do the data support the conclusions?

Reviewer #1: Partly

Reviewer #2: No

Reviewer #3: Yes

2. Has the statistical analysis been performed appropriately and rigorously? 

Reviewer #1: No

Reviewer #2: Yes

Reviewer #3: No

3. Have the authors made all data underlying the findings in their manuscript fully available?

Reviewer #1: Yes

Reviewer #2: No

Reviewer #3: Yes

4. Is the manuscript presented in an intelligible fashion and written in standard English?

Reviewer #1: Yes

Reviewer #2: No

Reviewer #3: Yes

5. Review Comments to the Author

Reviewer #1: Thank you for submitting your manuscript for consideration. Your topic choice is an excellent opportunity for a systematic review. However, I have identified several areas where improvements are necessary to meet the publication standards:

Abstract Structure: The abstract currently lacks an introduction section. I also noticed that certain abbreviations, such as "PFP," are used in the abstract without prior definition.

Literature Search: The literature search appears to be limited in scope. For a comprehensive review, it is essential to include major databases such as COCHRANE and Scopus. for example wondering why this important study is not included? https://pubmed.ncbi.nlm.nih.gov/33927114/

Presentation of Findings: The major findings of the review are not summarized in Table 1. Consider creating an additional table (table 2) that shows the effects of the intervention in each study. This table should ideally compare the differences in outcome measures from baseline, which will enhance the clarity and impact of your results.

Formatting: The references are not formatted according to the PLOS ONE style guide.

Figures: Titles are missing for the plots and figures, and their captions are incorrectly placed in the manuscript texts instead of below the figures. Additionally, to enhance the readability and accessibility of your table, I recommend numbering each abbreviation mentioned within the tables and providing a keyed list of these abbreviations below the table.

Reviewer #2: Introduction

The article cited to support the phrase "Additionally, muscle power is considered a better predictor of muscle function than strength (7)" refers to a review whose findings do not provide consistent evidence to support this statement.

It is essential for the authors to clearly state the gap in the field of knowledge that the manuscript will address. The studies cited to suggest controversies in the literature (12, 14, and 15) utilized different types of exercises and equipment; therefore, they do not necessarily present opposing results but rather different responses to varying exercise doses.

It is imperative to clarify in the introduction the meaning of "physical functional performance" as the set of tests indicated in the method to assess the primary outcome (mobility tests such as Five Times Sit-To-Stand, 30-sec chair stand, Backward walking, Timed Up and Go (TUG), Timed Up, and Go dual-task, and the Short Physical Performance Battery) measure distinct parameters. The study's objective is clearly defined.

Method

Participants: In the inclusion criteria, why did the authors include studies with Parkinson's patients in different stages of the disease? Is the response to an exercise-based intervention not influenced by the disease stage?

Intervention: The intervention description conflicts with the protocol registered in Prospero (Protocol number CRD42019139446). The types of exercises and equipment used for training differed among the included studies, as well as the muscle groups involved.

Control: The use of comparators, control without intervention, and the control being other forms of physical activity or exercise (e.g., balance training, resistance training, aerobic exercise, Tai Chi, movement strategy training, fall prevention education, and home exercise prescription) could potentially introduce a confounding factor, as the speed of exercise execution is not consistently specified in the method descriptions of the articles included in this SR. In the study by Meng Ni et al. (2016), the control group had training characteristics resembling power training. The authors demonstrated that High-Speed yoga improves power similarly to Power Training. Could this study not be a confounding factor considering the objective of the systematic review under analysis? The study by Meng Ni et al. (2016) is referenced throughout the manuscript as being from 2015.

Outcome: Authors must clearly define what was considered Functional Physical Performance (DFP), since the protocols listed for the primary outcome do not represent, in isolation, functional capacity.

Results

Considering the characteristics of the RCT interventions included in this SR, I understand that a meta-analysis would not be appropriate.

Why did the authors not use the Risk of Bias 2 (ROB 2) risk of bias analysis tool, which is better suited for SR’s of RCT’s that are not physical therapy interventions.

Discussion

The authors note that "The results of the present study may partially be explained by some factors. First, some studies showed high heterogeneity – ranging from 88% for PWS and 83% for FWS". These two parameters among the seven analyzed show a high coefficient of heterogeneity; for the other studies, there was no heterogeneity (i2 = 0). Additionally, the authors highlight a range of factors that may explain the analysis results, such as variations in the training protocols of the included studies and the involved muscle groups.

Why analyze studies with such diverse exercise doses? The discussion of the results of this SR is compromised due to the various limitations that compromise the meta-analysis of the data. Such limitations are presented by the authors in the discussion section.

The discussion is more focused on explaining the factors that compromise the meta-analysis than its potential results. This is because there is no room for a meta-analysis considering the studies included.

Reviewer #3: I am grateful for the opportunity to provide feedback on the manuscript entitled "EFFECT OF POWER TRAINING ON PHYSICAL FUNCTIONAL PERFORMANCE OF PATIENTS WITH PARKINSON’S DISEASE: A SYSTEMATIC REVIEW AND METAANALYSIS OF RANDOMIZED CONTROLLED TRIALS" submitted to PLOS One. Overall, the manuscript is well-crafted; however, several points warrant consideration.

1- Please elucidate the concept of power training within the "Intervention" component of your Population, Intervention, Comparison, Outcome (PICO) framework. Specifically, clarify the types of exercises categorized as power training within the context of your systematic review.

2- The rationale behind the authors' selection of Hedges' g over Cohen's d as the preferred effect size measure is not evident. Justification for this choice is necessary to enhance the clarity and rigor of the methodology.

3- I recommend employing the Cochrane Version 2 Risk of Bias (RoB 2) tool instead of the PEDro scale. The RoB 2 tool is specifically designed for assessing the risk of bias in interventional studies, aligning more closely with the objectives of this review.

4- Given the systematic nature of this review involving interventional studies, it is imperative for the authors to evaluate the certainty of evidence using the Grading of Recommendations Assessment, Development, and Evaluation (GRADE) approach.

5- The methodology utilized in the random effects model, such as the DerSimonian-Laird (DL) or Restricted Maximum Likelihood (REML) method, should be clearly specified.

6- In the forest plots, it is essential to denote which side favors power training and which side favors the control treatment.

7- Notably, there is a deficiency in the manuscript's discussion regarding potential mechanisms underlying considerable statistical heterogeneity observed among the results.

6. PLOS authors have the option to publish the peer review history of their article (what does this mean?). If published, this will include your full peer review and any attached files.

Reviewer #1: No

Reviewer #2: No

Reviewer #3: No

---

## [Author Response · Author response to Decision Letter 0]

15 Sep 2024

All responses to the reviewers' and editor's comments have been included in the document 'Response to Reviewers' attached in the previous submission step. Please refer to that document for detailed responses to each point raised.

---

## [Decision Letter · Decision Letter 1]

5 Nov 2024

EFFECT OF POWER TRAINING ON PHYSICAL FUNCTIONAL PERFORMANCE OF PATIENTS WITH PARKINSON’S DISEASE: A SYSTEMATIC REVIEW AND META-ANALYSIS OF RANDOMIZED CONTROLLED TRIALS

PONE-D-23-42955R1

Dear Dr. BRAGA-NETO,

We’re pleased to inform you that your manuscript has been judged scientifically suitable for publication and will be formally accepted for publication once it meets all outstanding technical requirements.

Kind regards,

Hamid Reza Baradaran, M.D., Ph.D.,

Academic Editor

PLOS ONE

Additional Editor Comments (optional):

Reviewers' comments:

Reviewer's Responses to Questions

**Comments to the Author**

1. If the authors have adequately addressed your comments raised in a previous round of review and you feel that this manuscript is now acceptable for publication, you may indicate that here to bypass the “Comments to the Author” section, enter your conflict of interest statement in the “Confidential to Editor” section, and submit your "Accept" recommendation.

Reviewer #3: All comments have been addressed

2. Is the manuscript technically sound, and do the data support the conclusions?

Reviewer #3: Yes

3. Has the statistical analysis been performed appropriately and rigorously? 

Reviewer #3: Yes

4. Have the authors made all data underlying the findings in their manuscript fully available?

Reviewer #3: Yes

5. Is the manuscript presented in an intelligible fashion and written in standard English?

Reviewer #3: Yes

6. Review Comments to the Author

Reviewer #3: Thank you for the opportunity to review the revised manuscript once again. The authors have put in considerable effort to address the reviewers' comments, and their responses meet my expectations. I have no further comments to provide.

7. PLOS authors have the option to publish the peer review history of their article (what does this mean?). If published, this will include your full peer review and any attached files.

Reviewer #3: No

---

## [Editor Report · Acceptance letter]

18 Dec 2024

PONE-D-23-42955R1 

PLOS ONE

Dear Dr. BRAGA-NETO, 

I'm pleased to inform you that your manuscript has been deemed suitable for publication in PLOS ONE. Congratulations! Your manuscript is now being handed over to our production team.

Kind regards, 

on behalf of

Professor Hamid Reza Baradaran 

Academic Editor

PLOS ONE